# Exploiting the mediating role of the metabolome to unravel transcript-to-phenotype associations

**Chiara Auwerx[1,2,3,4], Marie C Sadler[2,3,4], Tristan Woh[4], Alexandre Reymond[1], Zoltán Kutalik[2,3,4]\*, Eleonora Porcu[1,2,3]\***

[1]Center for Integrative Genomics, University of Lausanne, Lausanne, Switzerland; [2]Swiss Institute of Bioinformatics, Lausanne, Switzerland; [3]University Center for Primary Care and Public Health, Lausanne, Switzerland; [4]Department of Computational Biology, University of Lausanne, Lausanne, Switzerland

**\*For correspondence:**
eleonora.porcu@unil.ch (EP);
zoltan.kutalik@unil.ch (ZK)

**Competing interest:** The authors declare that no competing interests exist.

**Abstract** Despite the success of genome-wide association studies (GWASs) in identifying genetic variants associated with complex traits, understanding the mechanisms behind these statistical associations remains challenging. Several methods that integrate methylation, gene expression, and protein quantitative trait loci (QTLs) with GWAS data to determine their causal role in the path from genotype to phenotype have been proposed. Here, we developed and applied a multi-omics Mendelian randomization (MR) framework to study how metabolites mediate the effect of gene expression on complex traits. We identified 216 transcript-metabolite-trait causal triplets involving 26 medically relevant phenotypes. Among these associations, 58% were missed by classical transcriptome-wide MR, which only uses gene expression and GWAS data. This allowed the identification of biologically relevant pathways, such as between *ANKH* and calcium levels mediated by citrate levels and *SLC6A12* and serum creatinine through modulation of the levels of the renal osmolyte betaine. We show that the signals missed by transcriptome-wide MR are found, thanks to the increase in power conferred by integrating multiple omics layer. Simulation analyses show that with larger molecular QTL studies and in case of mediated effects, our multi-omics MR framework outperforms classical MR approaches designed to detect causal relationships between single molecular traits and complex phenotypes.

## Editor's evaluation

The reviewers found that your article brings important new methods and insight for how to analyze large, complex, multi-omic datasets in order to highlight specific molecular hypotheses for follow-up validation. As realistic/affordable sample sizes for population studies with omics data have recently exploded in size, this has raised the clear need for new or adapted statistical methods for best exploiting the increased statistical power. This work is compelling and we believe it should be of general interest to biologists and biostatisticians, and of particular interest to those working on (or those who could work on) large cohorts.

## Introduction

Genome-wide association studies (GWASs) have identified thousands of single nucleotide polymorphisms (SNPs) associated with a wide range of complex traits (*MacArthur et al., 2017*; *Visscher et al., 2017*). However, the path from GWAS to biology is not straightforward as most SNPs implicated by GWASs reside in non-coding regions of the genome (*MacArthur et al., 2017*) and do not directly inform on the functional mechanism through which variants exert their effect on phenotypes.

**Figure 1.** Workflow overview. (**A**) Estimation of the causal transcript-to-metabolite and metabolite-to-phenotype effects through univariable Mendelian randomization (MR). (**B**) Estimation of the causal transcript-to-phenotype effects through univariable transcriptome-wide MR (TWMR). (**C**) Estimation of the direct (i.e., not mediated by the metabolites) and mediated effect of transcripts on phenotypes through multivariable MR (MVMR) by accounting for mediation through the metabolome.

The online version of this article includes the following source data and figure supplement(s) for figure 1:

**Figure supplement 1.** Number of instrumental variables (IVs) used for causal effect estimation through Mendelian randomization (MR).

**Figure supplement 1—source data 1.** Number of instrumental variables (IVs) used for causal effect estimation through Mendelian randomization (MR).

GWASs have been performed on gene expression (*Võsa et al., 2021*), DNA methylation (*Min et al., 2021*), protein (*Sun et al., 2018*), and metabolite (*Shin et al., 2014*; *Lotta et al., 2021*) levels, identifying genetic variants influencing molecular traits, commonly referred to as molecular quantitative trait loci (molQTLs). The large overlap between complex and molecular trait-associated variants suggests that integrating these data can help interpreting GWAS loci (*Vandiedonck, 2018*; *Taylor et al., 2019*; *Ongen et al., 2017*). Advances in the field of transcriptomics make gene expression the best studied molecular phenotype, thanks to the presence of large expression QTL (eQTL) studies (e.g., eQTLGen Consortium [*Võsa et al., 2021*] N>30,000). Availability of these datasets fostered the development of summary statistic-based statistical approaches aiming at identifying associations between transcripts and complex traits (*Zhu et al., 2016*; *Porcu et al., 2019*; *Hormozdiari et al., 2014*; *Gusev et al., 2016*), prioritizing genes from known GWAS loci for functional follow-up, and inferring the directionality of these relations (*Porcu et al., 2019*; *Porcu et al., 2021b*). However, the cascade of events that mediates the effect of genetic variants on complex traits involves more than one molecular trait. Although approaches used for gene expression can be extended to other molecular data, investigating whether these molecular traits reside along the same causal pathway remains under-explored and only recently have studies applied colocalization and Mendelian randomization (MR) to methylation, gene expression, and protein levels data (*Giambartolomei et al., 2018*; *Wu et al., 2018*; *Gleason et al., 2020*; *Sadler et al., 2022*) and to a lesser extent to metabolic QTLs (mQTL) (*Yin et al., 2022*).

Metabolites are often the final products of cellular regulatory processes and the most proximal omic layer to complex phenotypes. Their levels could thus represent the ultimate response of biological systems to genetic and environmental changes. For instance, the metabolic status of organisms reflects disease progression, as metabolic disturbances can often be observed several years prior to the symptomatic phase (*Shah et al., 2010*; *Wang et al., 2011*; *Sabatine et al., 2005*). Therefore,

using metabolomics to identify early-stage biomarkers of complex phenotypes, such as prediabetes and COVID-19 susceptibility, has gained increased interest (*Wang-Sattler et al., 2012*; *Julkunen et al., 2021*). While two-sample MR approaches using metabolites as single exposure have revealed biomarkers for several diseases (*Qian et al., 2021*; *Lord et al., 2021*; *Porcu et al., 2021a*), these analyses focused on the prediction of disease risk rather than on deciphering the mechanisms of discovered associations.

Integrating transcriptomics with metabolomics data can provide insights into how metabolites are regulated, elucidating targetable functional mechanisms. Here, we develop a framework based on established MR methodology that hypothesizes a mediating role of the metabolome in the transcript-to-phenotype axis, with the primary exposure being defined as an upstream omic layer, namely gene expression. Specifically, our integrative MR analysis combines summary-level multi-omics (i.e., GWAS, eQTL, and mQTL) data to compute the indirect effect of gene expression on complex traits mediated by metabolites in three steps (*Figure 1*). First, we map the transcriptome to the metabolome by identifying causal associations between transcripts and metabolites. Next, we screen metabolites for downstream causal effects on 28 complex phenotypes, resulting in the identification of gene expression → metabolite → phenotype cascades (*Figure 1A*). In parallel, we prioritize trait-associated genes by testing the association of transcripts with phenotypes (*Figure 1B*). Third, for transcripts found to causally influence either a metabolite (A) or a complex phenotype (B), we test whether the identified target genes exert their effect on the phenotype through the metabolite using multivariable MR (MVMR; *Figure 1C*). Finally, we carried out extensive power analyses to determine under which conditions the mediation analysis (*Figure 1C*) outperforms the conventional exposure-outcome MR framework (*Figure 1B*).

## Results

### Mapping the transcriptome onto the metabolome

We applied univariable MR to identify metabolites whose levels are causally influenced by transcript levels in whole blood (*Figure 1A*). Summary statistics for *cis*-eQTLs stem from the eQTLGen Consortium meta-analysis of 19,942 transcripts in 31,684 individuals (*Võsa et al., 2021*), while summary statistics for mQTLs originate from a meta-analysis of 453 metabolites in 7824 individuals from two independent European cohorts: TwinsUK (*N* = 6056) and KORA (*N* = 1768) (*Shin et al., 2014*). After selecting SNPs included in both the eQTL and mQTL studies, our analysis was restricted to 7884 transcripts with ≥3 instrumental variables (IVs) (see Methods, *Figure 1—figure supplement 1A*) and 242 metabolites with an identifier in the Human Metabolome Database (HMDB) (*Wishart et al., 2022*) (see Methods, *Supplementary file 1a*). By testing each gene for association with the 242 metabolites, we detected 96 genes whose transcript levels causally impacted 75 metabolites, resulting in 133 unique transcript-metabolite associations (FDR 5% considering all 1,907,690 instrumentable gene-metabolite pairs; *Supplementary file 1b*). Most involved genes (86%; 83/96) were causally influencing the level of a single metabolite, with some notable exceptions acting as mQTL hubs, such as *TMEM258* and *FADS2* both affecting the same 11 metabolites, followed by *FADS1* affecting a subset of six metabolites. While only 5 (3.8%) of the 133 associations were reported in HMDB, an automated literature review (see Methods) identified a match for 22 (16.5%) of the identified transcript-metabolite pairs (*Supplementary file 1b*).

### Mapping the metabolome onto complex phenotypes

Univariable metabolome-wide MR (MWMR; *Figure 1A*) was used to identify causal relationships between 48 metabolites with ≥3 IVs (*Figure 1—figure supplement 1B*) and 28 complex phenotypes. The latter include a wide range of anthropometric traits, cardiovascular assessments, and blood biomarkers, whose summary statistics originate from the UK Biobank (UKB) (*Bycroft et al., 2018*; *Supplementary file 1c*). Overall, 34 metabolites were associated with at least one phenotype (FDR 5% considering all 1344 metabolite-phenotype pairs), resulting in 132 unique metabolite-phenotype associations (*Supplementary file 1d*).

### Mapping the transcriptome onto complex phenotypes

We applied univariable transcriptome-wide MR (TWMR [*Porcu et al., 2019*] *Figure 1B*) to identify associations between expression levels of 10,435 transcripts from the eQTLGen Consortium with ≥3 IVs

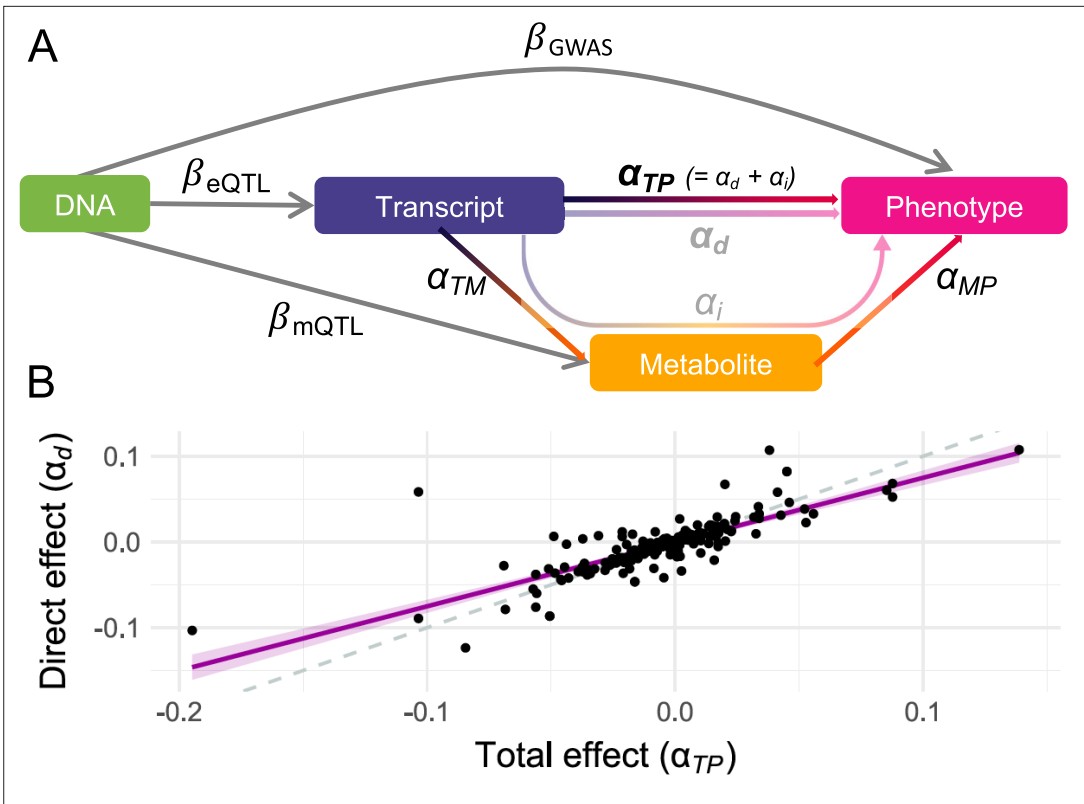

**Figure 2.** Direct and mediated effects. (**A**) Graphical representation of the multivariable Mendelian randomization (MVMR) framework for mediation analysis: DNA represents genetic instrumental variables (IVs) chosen to be directly associated with either the exposure (transcript; $\beta_{eQTL}$) or the mediator (metabolite; $\beta_{mQTL}$) through summary statistics. The effect of these IVs on the outcome (phenotype; $\beta_{GWAS}$) originates from genome-wide association studies (GWASs) summary statistics. Total effects $\alpha_{TP}$ of transcripts on phenotypes are estimated by transcriptome-wide Mendelian randomization (TWMR), while direct effects $\alpha_d$ are estimated by MVMR. Total effects $\alpha_{TP}$ are assumed to equal the sum of the direct $\alpha_d$ and indirect $\alpha_i$ (i.e., mediated) effects, the two former being depicted in **B**. (**B**) Direct ($\alpha_d$ ; y-axis) and total ($\alpha_{TP}$ ; x-axis) effects for the 216 transcript-metabolite-trait causal triplets. The dashed line represents the identity, while the purple line represents the regression line with a shaded 95% confidence interval. Data related to *Figure 2* panel B are available in *Figure 2—source data 1*.

The online version of this article includes the following source data for figure 2:

**Source data 1.** Direct and mediated effects.

(*Figure 1—figure supplement 1C*) measured in both exposure and outcome datasets and the same 28 UKB phenotypes described in the previous section (*Supplementary file 1c*). In total, 5140 transcripts associated with at least one phenotype (FDR 5% considering all 292,170 gene-phenotype pairs) resulting in 13,141 unique transcript-phenotype associations (*Supplementary file 1e*).

## Mapping metabolome-mediated effects of the transcriptome onto complex phenotypes

The mapping of putative causal effects performed in the previous steps provides the opportunity to infer the mediating role of the metabolome in biological processes leading to transcript-phenotype associations. We combined the 133 transcript-metabolite (FDR ≤5%) and 132 metabolite-trait (FDR ≤5%) associations to pinpoint 216 transcript-metabolite-phenotype causal triplets (FDR = 1–0.95² = 9.75%) (*Supplementary file 1f*). Among the 37 triplets for which the transcript and metabolite had previously been linked through automated literature review, none remained after incorporating a third term for the phenotype in the search and manually removing abstracts for which the search terms were used in an erroneous context. Relaxing the search criteria by omitting the metabolite term, 13/37 (35%) triplets returned at least one match for the gene-trait association.

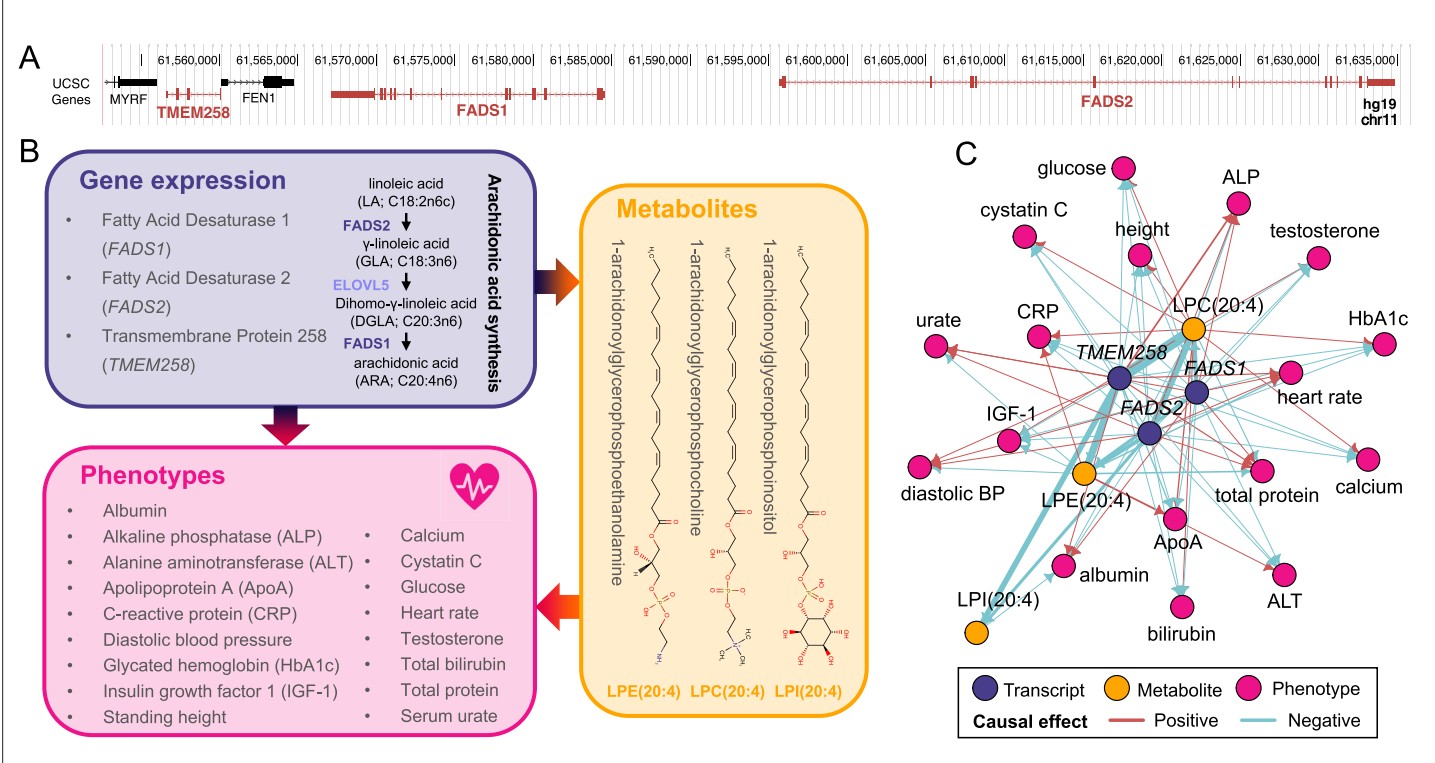

**Figure 3.** Molecular pleiotropy at the *FADS* locus. (**A**) Genome browser (GRCh37/hg19) view of the genomic region on chromosome 11 encompassing *TMEM258*, *FADS1*, and *FADS2* (red). (**B**) Diagram of the mediation signals detected for *TMEM258*, *FADS1*, and *FADS2*. Two of the implicated genes encode enzymes involved in arachidonic synthesis (purple). Involved genes impact 17 phenotypes (pink) through alteration of the levels of three metabolites, 1-arachidonoylglycerophosphocholine (LPC(20:4)), 1-arachidonoylglycerophosphoethanolamine (LPE(20:4)), and 1-arachidonoylglyceropho sphoinositol (LPI(20:4)) whose structure is depicted (orange). (**C**) Network of the 65 transcript-metabolite-trait causal triplets involving *TMEM258*, *FADS1*, and *FADS2*. Nodes represent genes (purple), metabolites (orange), or phenotypes (pink). Edges indicate the direction of the effects estimated through univariable Mendelian randomization. Width of edges is proportional to effect size and color indicates if the effect is positive (red) or negative (blue).

For each of these 216 putative mechanisms, an MVMR approach to compute the direct effect of gene expression on the phenotype was applied (see Methods; *Figure 1C*; *Supplementary file 1f*). Regressing direct effects ($\alpha_d$) on total effects ($\alpha_{TP}$) and accounting for regression dilution bias (see Methods; *Figure 2A*), it was estimated that 77% [95% CI: 70–85%] of the transcript effect on the phenotype was direct and thus not mediated by the metabolites (*Figure 2B*).

## Molecular mechanisms of genotype-to-phenotype associations

Dissecting causal triplets allows gaining mechanistc insights into biological pathways linking genes to phenotypes. For instance, expression of *TMEM258* [MIM: 617615], *FADS1* [MIM: 606148], and *FADS2* [MIM: 606149], all mapping to a region on chromosome 11 (*Figure 3A*), were found to influence a total of 17 complex phenotypes through modulation of 1-arachidonoylglycerophosphochol ine (LPC(20:4); HMDB0010395; $\alpha_{TMEM258 \rightarrow LPC(20:4)} = -1.02$; P = 8.0×10⁻⁸¹; $\alpha_{FADS1 \rightarrow LPC(20:4)} = -0.39$; P = 4.6×10⁻¹⁵; $\alpha_{FADS2 \rightarrow LPC(20:4)} = -0.63$; P = 5.1×10⁻⁶²), 1-arachidonoylglycerophosphoethanolamine (LPE(20:4);HMDB0011517; $\alpha_{TMEM258 \rightarrow LPE(20:4)} = -0.68$; P = 1.1×10⁻³⁷; $\alpha_{FADS1 \rightarrow LPE(20:4)} = -0.30$; P = 1.4×10⁻⁰⁷; $\alpha_{FADS2 \rightarrow LPE(20:4)} = -0.37$; P = 1.2×10⁻¹⁸), and 1-arachidonoylglycerophosphoinositol (LPI(20:4); HMDB0061690; $\alpha_{TMEM258 \rightarrow LPI(20:4)} = -0.51$; P = 8.2×10⁻¹⁸; $\alpha_{FADS2 \rightarrow LPI(20:4)} = -0.28$; P = 6.3×10⁻¹⁶) levels (*Figure 3B–C*). These results align with the known pleiotropy of the region (i.e., >6000 associations reported in the GWAS Catalog as of May 2022). Interestingly, involved metabolites are complex lipids synthesized from arachidonic acid, a product of the rate-limiting enzymes encoded by *FADS1* and *FADS2* (*Figure 3B*). Recently, polymorphisms affecting the expression of these genes were shown to associate with the levels of over 50 complex lipids, including the ones identified by our study

(*Reynolds et al., 2020*). Overall, this example illustrates how our method can capture meaningful biological associations and shed light on underlying molecular pathways of pleiotropy.

## Power analysis

Importantly, only 42% (90/216) of the causal triplets showed a significant total transcript-to-phenotype effect (i.e., estimated by TWMR), suggesting that the method lacks power under current settings. To characterize the parameter regime where the power to detect indirect effects is larger than it is for total effects, we performed simulations using different settings for the mediated effect. In each scenario we evaluated 500 transcripts and 80 metabolites and varied two parameters characterizing the mediation:

1. The proportion ($\rho$) of direct ($\alpha_d$) to total ($\alpha_{TP}$) effect (i.e., effect not mediated by the metabolite) from –2 to 2 to cover the cases where direct and mediated effect have opposite directions (51 values).
2. The ratio ($\sigma$) between the transcript-to-metabolite ($\alpha_{TM}$) and the metabolite-to-phenotype ($\alpha_{MP}$) effects, exploring the range from 0.1 to 10 (51 values).

Transcripts were simulated with 6% heritability (i.e., median $h^2$ in the eQTLGen data) and a causal effect of 0.035 (i.e., ~65% of power in TWMR at $\alpha=0.05$) on a phenotype. Each scenario was simulated 10 times and results were averaged to assess the mean difference in power (see Methods).

Simulations show that with current sample sizes (i.e., $N_{GWAS} = 300,000$, $N_{eQTL} = 32,000$, and $N_{mQTL} = 8000$), when $\alpha_{MP} > \alpha_{TM}$ (i.e., $\sigma < 1$), TWMR has increased power to detect significant transcript-to-phenotype associations, especially when $\rho > 0$ (i.e., direct and total effect have the same direction (*Figure 4A*)). However, for all 216 causal triplets, we observed $\sigma > 1$ (*Figure 4—figure supplement 1*). Under this condition, and assuming that the total effect of the transcript on the phenotype is dominated by the effect mediated by the metabolite (i.e., $\rho < 0.5$ and $\rho > 1.5$), TWMR had less power than the approach identifying mediators (*Figure 4A*), confirming that significant associations were missed by TWMR due to power issues related to the proportion of mediated effect.

Repeating the simulations with an mQTL sample size of 90,000, nearing state-of-the-art sample sizes (*Lotta et al., 2021*), leads to a strong shift in the above-described trends (*Figure 4B*). Specifically, when the effect of the transcript on the phenotype is dominated by the effect mediated by the metabolite ($\rho < 0.3$ and $\rho > 1.7$), mediation analysis has more power than TWMR when $\sigma > 0.2$. For larger proportions of direct effect, TWMR has increased power the more $\sigma$ differs from 1. In line with the increased power of mediation analysis with larger mQTL datasets, the gain in power of mediation analysis over TWMR decreases with decreasing mQTL dataset sample sizes (ranging between N = 1000 and N = 4000; *Figure 4—figure supplement 2*), indicating that our approach is dependent on large sample sizes to reach its full potential.

## Identifying new genotype-to-phenotype associations

The 126 triplets that were not identified through TWMR due to power issues represent putative new causal relations. This is well illustrated by a proof-of concept example involving *ANKH* [MIM: 605145] and calcium levels, for which 48 publications were identified through automated literature review (*Supplementary file 1f*). While the TWMR effect of *ANKH* expression on calcium levels was not significant ($\alpha_{ANKH \to calcium} = -0.02$; P=0.03), *ANKH* expression decreased citrate levels ($\alpha_{ANKH \to citrate} = -0.30$; P=2.2×10$^{-06}$), which itself increased serum calcium levels ($\alpha_{citrate \to calcium} = 0.07$; P=6.5×10$^{-0}$). Mutations in *ANKH* have been associated with several rare mineralization disorders [MIM: 123000, 118600] (*Williams, 2016*) due to the gene encoding a transmembrane protein that channels inorganic pyrophosphate to the extracellular matrix, where at low concentrations it inhibits mineralization (*Ho et al., 2000*). Recently, a study proposed that ANKH instead exports ATP to the extracellular space (where it is then rapidly converted to inorganic pyrophosphate), along with citrate (*Szeri et al., 2020*). Citrate has a high binding affinity for calcium and influences its bioavailability by complexing calcium-phosphate during extracellular matrix mineralization and releasing calcium during bone resorption (*Granchi et al., 2019*). Together, our data support the role of ANKH in calcium homeostasis through regulation of citrate levels, connecting previously established independent links into a causal triad.

In another example, *SLC6A12* [MIM: 603080], which encodes the betaine/GABA transporter-1, involved in betaine and GABA uptake (*Borden et al., 1995*), was identified as a negative regulator of betaine ($\alpha_{SLC6A12 \to betaine} = -0.37$; P = 8.2×10$^{-08}$). While blood betaine levels negatively impacted

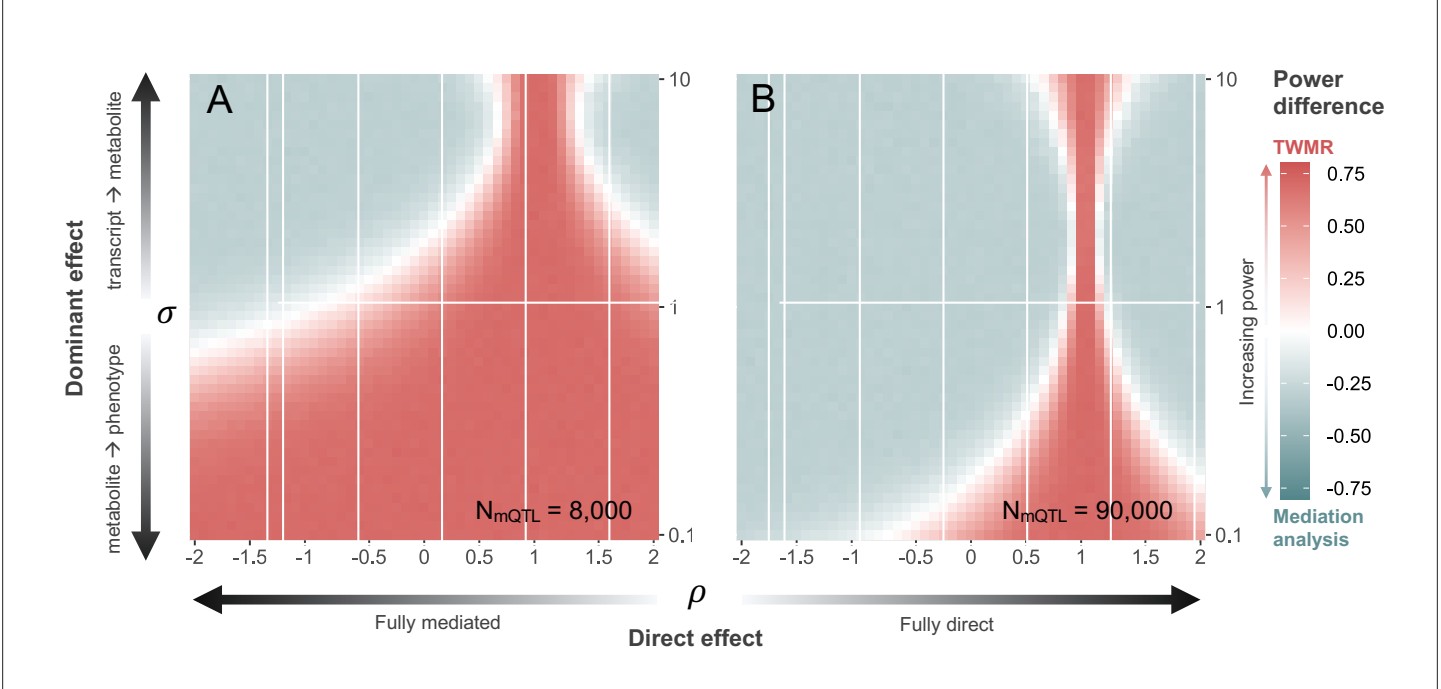

**Figure 4.** Power comparison between transcriptome-wide Mendelian randomization (TWMR) and multivariable Mendelian randomization (MVMR). Heatmap showing the difference in statistical power between TWMR and mediation analysis through MVMR at current (**A**; $N = 8000$) and realistic future (**B**; $N = 90,000$) metabolic quantitative trait loci (mQTL) dataset sample sizes. The x-axis shows the proportion ($\rho$) of direct ($\alpha_d$) to total ($\alpha_{TP}$) effect (i.e., effect not mediated by the metabolite) ranging from –2 to 2, arrows indicating increasing proportion of direct effect. The y-axis shows the ratio ($\sigma$) between the transcript-to-metabolite ($\alpha_{TM}$) and the metabolite-to-phenotype ($\alpha_{MP}$) effects, ranging from 0.1 to 10. Red vs. gray indicates higher power for TWMR vs. mediation analysis, respectively, while white represents equal power between the two approaches. Data related to *Figure 4* panels A and B are available in *Figure 4—source data 1* and *Figure 4—source data 2*, respectively.

The online version of this article includes the following source data and figure supplement(s) for figure 4:

**Source data 1.** Difference in statistical power between transcriptome-wide Mendelian randomization (TWMR) and mediation analysis at $N = 8000$ metabolic quantitative trait locus (mQTL) dataset sample size.

**Source data 2.** Difference in statistical power between transcriptome-wide Mendelian randomization (TWMR) and mediation analysis at $N = 90,000$ metabolic quantitative trait locus (mQTL) dataset sample size.

**Figure supplement 1.** Distribution of empirical causal triplets along tested regime parameters.

**Figure supplement 1—source data 1.** Distribution of empirical causal triplets along tested regime parameters.

**Figure supplement 2.** Power comparison between transcriptome-wide Mendelian randomization (TWMR) and multivariable Mendelian randomization (MVMR) at smaller sample sizes.

**Figure supplement 2—source data 1.** Difference in statistical power between transcriptome-wide Mendelian randomization (TWMR) and mediation analysis at $N = 1000$ metabolic quantitative trait locus (mQTL) dataset sample size.

**Figure supplement 2—source data 2.** Difference in statistical power between transcriptome-wide Mendelian randomization (TWMR) and mediation analysis at $N = 2000$ metabolic quantitative trait locus (mQTL) dataset sample size.

**Figure supplement 2—source data 3.** Difference in statistical power between transcriptome-wide Mendelian randomization (TWMR) and mediation analysis at $N = 4000$ metabolic quantitative trait locus (mQTL) dataset sample size.

serum creatinine levels ($\alpha_{betaine \rightarrow creatinine} = -0.06$; P = $1.7 \times 10^{-07}$), the effect of *SLC6A12* expression on creatinine was not significant ($\alpha_{SLC6A12 \rightarrow creatinine} = 0.02$; P = $1.5 \times 10^{-03}$). This observation is particularly interesting given that betaine acts as a protective renal osmolyte whose plasma and kidney tissue concentration were found to be downregulated in renal ischemia/reperfusion injury (*Jouret et al., 2016*; *Wei et al., 2014*) and whose urine levels have been proposed as a biomarker for chronic kidney disease progression (*Gil et al., 2018*). As both renal conditions are commonly monitored through serum creatinine levels, our data support the critical role of osmolyte homeostasis in renal health.

## Discussion

In this study, we combined MR approaches integrating eQTL, mQTL, and GWAS summary statistics to explore the role of the metabolome in mediating the effect of the transcriptome on complex phenotypes. Applied to 28 medically relevant traits, our approach revealed 216 causal transcript-metabolite-phenotype triplets. Our automated literature review indicates that while some detected associations were previously reported, a large fraction, especially among the triplets, appears to be novel. It should be noted that the number of previously reported associations is likely underestimated as our approach does not account for all synonyms of a given feature and requires the terms to appear in the title or abstract of the publication. This makes it more likely for hypothesis-driven studies, inherently biased toward well-studied genes and metabolites, to be identified. Conversely, high-throughput, hypothesis-free studies that report the given association in a supplemental table are likely to be missed. Furthermore, due to its automated nature, our search is context-blind, so that some of the identified studies might report negative results, associations only under specific conditions (e.g., different organisms, experimental settings), or usage of the search term with a different meaning. To attenuate the latter, we also performed manual review of the retained abstracts for transcript-to-phenotype searches. While flawed, this rough estimate of the amount of existing evidence supporting our findings can be interpreted in combination with other lines of evidence. For instance, among the 90 signals that were also identified through TWMR, 93% showed a directionally concordant effect between the transcript-to-phenotype, transcript-to-metabolite, and metabolite-to-phenotype estimates (i.e., sign of product of the transcript-to-metabolite and metabolite-to-phenotype effects agrees with the sign of the transcript-to-phenotype effect). In these situations, dissection of causal effects provides clues as to the molecular mechanism through which involved genes modify complex phenotypes. This information is particularly valuable to identify key molecular mediators of highly pleiotropic genetic regions, such as the *TMEM258/FADS1/FADS2* locus (*Figure 3*). While transcript levels of these genes affected eleven metabolites, three complex lipids were highlighted as strong molecular mediators of the transcript-to-phenotype effects.

Strikingly, 58% of the 216 causal transcript-metabolite-phenotype triplets were missed by TWMR – an approach that only considers gene expression and GWAS data. We highlight two novel but biologically plausible mechanisms linking *ANKH* to calcium levels through modulation of citrate and *SLC6A12* to serum creatinine levels through regulation of the renal osmolyte betaine. Simulation analyses showed that these signals were likely missed by TWMR due to lack of power, as mediation analysis is better suited to detect associations with a low direct to total effect proportion and stronger transcript-to-metabolite than metabolite-to-phenotype effect. Promisingly, our simulations showed that mediation analysis becomes increasingly powerful over a wider range of parameter settings as the sample size of the mediator QTL study increases, highlighting the importance of generating large and publicly available molQTL datasets that can help to unravel functional gene-to-phenotype mechanisms.

As illustrated through the selected examples, a large fraction of detected mediations involves genes encoding metabolic enzymes or transporters/channels, with an enrichment for 'secondary active transmembrane transporter activity', for example (GO:0015291; FDR=0.021; background: 7884 genes with ≥3 IVs assessed through TWMR; STRING database). Matching the finding that the most likely effector genes of mQTLs are enriched for pathway-relevant enzymes and transporters (*Smith et al., 2022*), these results are not surprising given that the proteins encoded by these genes directly interact with metabolites, making it more likely that the effect of changes in their expression is mediated by metabolites. While our method is well suited to detect such effects, interpretation of discovered mediations is limited by the lack of spatial resolution of the mQTL data. Access to metabolite concentrations in different cellular compartments (e.g., extracellular space, cytosol, mitochondrial matrix, etc.) would generate more fine-tuned mechanistic hypotheses that consider the directionality of metabolite fluxes.

The observation that 77% of the transcript's effect on the phenotype is not mediated by metabolites suggests that either true direct effects are frequent or that other unassessed metabolites or molecular layers (e.g., proteins, post-translational modifications, etc.) play a crucial role in mediation. It is to note that in the presence of unmeasured mediators or measured mediators without genetic instruments, our mediation estimates are lower bounds of the total existing mediation. In addition, unmeasured mediators sharing genetic instruments with the measured ones can modify result interpretation as

some of the observed mediators may simply be correlates of the true underlying mediators. While this is a limitation of all MR methods, metabolic networks may harbor particularly large number of genetically correlated metabolite species. Similarly, owing to linkage disequilibrium and regulatory variants affecting multiple genes, transcripts from adjacent genes might appear to be involved in the same signals, as exemplified with the *TMEM258/FADS1/FADS2* locus (*Figure 3*). While literature supports the role of the *FADS* genes, one cannot exclude a role for *TMEM258*, nor disentangle the specific function of *FADS1* and *FADS2*. Thanks to the flexibility of the proposed framework, we expect that in the future and upon availability of ever larger and more diverse datasets, our method could be applied to estimate the relative contribution of currently unassessed mediators in translating genotypic cascades.

Another consideration is that complex phenotypes can have a stronger impact on gene expression than the opposite (*Porcu et al., 2021b*). Due to the lack of genome-wide *trans*-eQTL association summary statistics, our method does not investigate reverse causality on metabolites and gene expression, nor the role of metabolites as regulators of gene expression. Metabolites might also integrate the effect of several transcripts (i.e., multiple transcripts causally impact the levels of the same metabolite) before affecting complex phenotypes (*Supplementary file 1g*) or multiple metabolites may jointly mediate the impact of a single transcript. Modelling the latter phenomenon, which is beyond the scope of our current work, requires the development of structural equation models accounting for such effects and will eventually lead to a more comprehensive modelling of causal relations in complex biological networks, nuancing the interpretation of the molecular mechanisms shaping complex traits.

In conclusion, we developed a modular MR framework that has increased power over classical MR approaches to detect causal transcript-to-phenotype relationships when these are mediated by alteration of metabolite levels and is likely to become increasingly powerful upon release of larger molQTL datasets.

## Methods
### Univariable MR analyses

TWMR and MWMR (*Porcu et al., 2019*) were used to estimate the causal effects of transcript and metabolite levels (exposure) on various outcomes. For each transcript/metabolite, using inverse-variance weighted (IVW) method for summary statistics (*Burgess et al., 2013*), the causal effect of the molecular traits on the outcome was defined as

$$\hat{\alpha} = \left( \beta' C^{-1} \beta \right)^{-1} \left( \beta' C^{-1} \gamma \right) \qquad (1)$$

Here, $\beta$ is a vector of length $n$ containing the standardized effect size of $n$ independent SNPs on the gene/metabolite, derived from eQTL/mQTL studies, with $\beta'$ being the transpose of $\beta$. $\gamma$ is a vector of length $n$ containing the standardized effect size of each SNP on the outcome. $C$ is the pairwise LD matrix between the $n$ SNPs. The standardized effect sizes for molecular and outcome GWASs were obtained from Z-score of summary statistics standardized by the square root of the sample size to be on the same standard deviation scale.

IVs were selected as autosomal, non-strand ambiguous, independent ($r^2$ <0.01), and significant ($P_{eQTL} < 1.8 \times 10^{-05}$ / $P_{mQTL} < 1.0 \times 10^{-07}$) eQTL/mQTLs available in the UK10K reference panel (*Huang et al., 2015*) using PLINK (v1.9) (*Chang et al., 2015*). As retained SNPs are independent, we used the identity matrix to approximate $C$. SNPs with larger effects on the outcome than on the exposure were removed, as these potentially indicate violation of the MR assumptions (i.e., likely reverse causality and/or confounding).

The variance of $\alpha$ can be calculated approximately by the Delta method

$$var\left(\hat{\alpha}\right) = \left(\frac{\partial \hat{\alpha}}{\partial \hat{\beta}}\right)^2 * var\left(\hat{\beta}\right) + \left(\frac{\partial \hat{\alpha}}{\partial \hat{\gamma}}\right)^2 * var\left(\hat{\gamma}\right) + \left(\frac{\partial \hat{\alpha}}{\partial \hat{\beta}}\right) * \left(\frac{\partial \hat{\alpha}}{\partial \hat{\gamma}}\right) * cov\left(\hat{\beta}, \hat{\gamma}\right) \qquad (2)$$

where $cov(\beta,\gamma)$ is 0 if $\beta$ and $\gamma$ are estimated from independent samples. The causal effect Z-statistic for transcript/metabolite $i$ was defined as $\frac{\hat{\alpha}_i}{SE(\hat{\alpha}_i)}$, where $SE\left(\hat{\alpha}_i\right) = \sqrt{var\left(\hat{\alpha}\right)_{i,i}}$ .

The IVW method provides an unbiased estimate under the assumption that all genetic variants are valid IVs, that is, all three MR assumption hold. However, the third assumption (no pleiotropy) is easily violated, leading to inaccurate estimates when horizontal pleiotropy occurs (**Verbanck et al., 2018**). To test for the presence of pleiotropy, we used Cochran's Q test (**Bowden et al., 2015**; **Burgess et al., 2017**) to assess whether there were significant differences between the MR effects of an instrument (i.e., $\alpha\beta_i$) and the estimated effect of that instrument on phenotype/metabolite levels ($\gamma_i$). We defined

$$d_i = \gamma_i - \alpha\beta_i \tag{3}$$

and its variance as

$$var\left(d_i\right) = var\left(\gamma_i\right) + \left(\beta_i\right)^2 * var\left(\alpha\right) + var\left(\gamma_i\right) * \left(\alpha\right)^2 + var\left(\beta_i\right) * var\left(\alpha\right) \tag{4}$$

Next, the deviation of each SNP was tested using the test statistic

$$T_i = \frac{d_i^2}{var(d_i)} \sim \chi_1^2 \tag{5}$$

When p<0.05, the SNP with largest $|d_i|$ was removed and the test was repeated.

## Mediation analysis through MVMR analyses

An MVMR approach was used to dissect the total causal effect of transcript levels on phenotypes ($\alpha_{TP}$) into a direct ($\alpha_d$) and indirect ($\alpha_i$) effect measured through a metabolite. Through inclusion of a metabolite and its associated genetic variants ($r^2$ <0.01, $p_{mQTL}$<1 × 10$^{-07}$), the direct effect of gene expression on a phenotype can be estimated using a multivariable regression model (**Burgess et al., 2013**) as the first element of

$$\hat{\alpha} = \left(B' C^{-1} B\right)^{-1} \left(B' C^{-1} \gamma\right) \tag{6}$$

where $B$ is a matrix with two columns containing the standardized effect sizes of $n$ IVs on transcript levels in the first column and on the metabolite levels in the second column, $\gamma$ is a vector of length $n$ containing the standardized effect size of each SNP on the phenotype, and $C$ is the pairwise LD matrix between the $n$ SNPs.

The proportion of direct effect ($\rho$) is calculated by regressing direct effects ($\alpha_d$) on total effects ($\alpha_{TP}$) and then correcting for regression dilution bias:

$$\rho_{corrected} = \frac{\rho}{\sqrt{1 - \frac{\sum(SE(\alpha_{TP}))^2}{\sum \alpha_{TP}^2}}} \tag{7}$$

## Omics and traits summary statistics

Expression QTL data originated from the eQTLGen Consortium (**Võsa et al., 2021**) (*N* = 31,684), which includes *cis*-eQTLs (<1 Mb from gene center, two-cohort filter) for 19,250 transcripts (16,934 with at least one significant *cis*-eQTL at FDR <0.05 corresponding to p<1.8 × 10$^{-05}$). mQTL data originate from **Shin et al., 2014**, which used ultra-high performance liquid chromatography-tandem mass spectrometry to measure 486 whole blood metabolites in 7824 European individuals. Association analyses were carried out on ~2.1 million SNPs and are available for 453 metabolites at the Metabolomics GWAS Server (http://metabolomics.helmholtz-muenchen.de/gwas/). Among these metabolites, 242 were manually annotated with the HMDB identifiers (**Supplementary file 1a**) and used in this study. GWAS summary statistics for 28 outcome traits measured in the UKB originate from the Neale Lab (http://www.nealelab.is/uk-biobank/). Protein interactions with metabolites were downloaded from HMDB v5.0 (https://hmdb.ca/downloads/) and were used to annotate transcript-metabolites associations.

## Automated literature review

An automated literature review of all transcript-metabolite associations (*Supplementary file 1b*) was conducted in PubMed (September 20, 2022) following the scheme:

$$\left(Gene\left[Title/Abstract\right]\right)\ AND\ \left(\left(Met_{HMDB}\left[Title/Abstract\right]\right)\ OR\ \left(Met_{Shin}\left[Title/Abstract\right]\right)\right) \tag{8}$$

With *Gene* being the name of the gene whose transcript is involved in the association, $Met_{HMDB}$ the involved metabolite's common name on HMDB, and $Met_{Shin}$ the involved metabolite's name as reported in *Shin et al., 2014*. Returned PubMed identifiers were retrieved (*Supplementary file 1b*).

For transcript-metabolite associations involved in a causal triplet and for which the transcript-metabolites returned at least one publication (*Supplementary file 1f*), the search was extend by (i) adding an additional search term for the trait (i.e., $AND\ \left(trait\left[Title/Abstract\right]\right)$) and (ii) substituting the metabolite term for the trait term. Returned PubMed identifiers were retrieved and corresponding abstracts were manually curated to exclude abstracts in which the search terms were used in a meaning other than the intended one (*Supplementary file 1f*).

## Simulation analyses

Simulation analyses were conducted to assess the gain in power upon inclusion of metabolomics data in the MR framework. In the simulated scenario, a transcript has an effect on a phenotype mediated by a metabolite. Two parameters were allowed to vary: the proportion ($\rho$) of direct effect (i.e., effect not mediated by the metabolite) and the ratio ($\sigma$) between the effect of the transcript on the metabolite ($\alpha_{TM}$) and of the metabolite on the phenotype ($\alpha_{MP}$). Other parameters were fixed, including the heritability of the transcript at $h_T^2 = 0.06$ (corresponding to the median $h^2$ in the eQTLGen data), the number of IVs $N_{IVs}$ at 6 (corresponding to the median number of IVs used in TWMR analyses). Effect sizes $\beta_{eQTL}$ are from a normal distribution $\beta_{eQTL} \sim N\left(0, \frac{h_T^2}{N_{IVs}}\right)$. The causal effect of the transcript on the phenotype ($\alpha_{TP}$) was fixed to 0.035, which results in ~65% power to detect a significant effect with TWMR. These quantities allowed to define $\beta_{GWAS}$ as $\beta_{GWAS} = \alpha_{TP} * \beta_{eQTL} + \varepsilon_P$, where $\varepsilon_P \sim N\left(0, \frac{1}{N_{GWAS}}\right)$ with $N_{GWAS} = 300,000$ to reflect the sample size of UKB GWASs. The same vector of $\beta_{eQTL}$ was used to define $\beta_{mQTL}$ and estimate the causal effect of the transcript on the metabolite. $\beta_{mQTL}$ was defined as $\beta_{mQTL} = \alpha_{TM} * \beta_{eQTL} + \varepsilon_M$, where $\varepsilon_M \sim N\left(0, \frac{1}{N_{mQTL}}\right)$ and $N_{mQTL} = 8000$ to reflect the sample size of the mQTL study used in this work. Simulations were also performed at $N_{mQTL} = 90,000$, to reflect sample size of potential future studies and $N_{mQTL} = 1000$, $N_{mQTL} = 2000$ and $N_{mQTL} = 4000$, to compare the two approaches' power were the developed framework to be applied on existing smaller mQTL datasets. The total effect $\alpha_{TP}$ can be expressed as $\alpha_{TP} = \alpha_{TM} * \alpha_{MP} + \alpha_{direct}$, where $\alpha_{direct}$ represents the direct effect of the transcript on the phenotype and $\alpha_{TM} * \alpha_{MP}$ is the indirect effect mediated by the metabolite. Equivalently, $\alpha_{TM} * \alpha_{MP} = \alpha_{TP} * (1 - \rho)$ where $\rho = \frac{\alpha_{direct}}{\alpha_{TP}}$. To assess the ratio between the effect of the transcript on the metabolite and the effect of the metabolite on the phenotype (i.e., $\sigma = \alpha_{TM}/\alpha_{MP}$), $\alpha_{TM}$ can be expressed as $\alpha_{TM} = \sqrt{\alpha_{TP} * (1 - \rho) * \sigma}$. Similarly, to estimate the effect of the metabolite on the phenotype, a metabolite with heritability $h_M^2 = 0.04$ (corresponding to the median of $h^2$ in the KORA +TwinsUK mQTL data) and $N_{IVs} = 5$ (corresponding to the median number of IVs used in MWMR analyses) is considered. Effect size $\beta_{mQTL}$ are from a normal distribution $\beta_{mQTL} \sim N\left(0, \frac{h_M^2}{N_{IVs}}\right)$. These quantities allowed to define $\beta_{GWAS}$ as $\beta_{GWAS} = \sqrt{\alpha_{TP} * (1 - \rho)/\sigma} * \beta_{mQTL} + \varepsilon_P$, where $\varepsilon_P \sim N\left(0, \frac{1}{N_{GWAS}}\right)$. Ranging $\rho$ and $\sigma$ from –2 to 2 and from 0.1 and 10, respectively, we run each simulation for 500 transcripts measuring 80 metabolites at each run and performed TWMR and MWMR starting from above-described $\beta_{eQTL}$, $\beta_{mQTL}$, and $\beta_{GWAS}$. For each MR analysis the power to detect a significant association as well as the difference in power between TWMR and the mediation analyses (i.e., $power_{TP} - power_{TM} * power_{MP}$) was calculated. Each specific scenario was repeated 10 times and the average difference in power across simulation was plotted as a heatmap.

## Data and code availability

All data used in this study are publicly available. GWAS summary statistics for outcome traits measured in the UKB originate from the Neale Lab (http://www.nealelab.is/uk-biobank/). eQTL data originated from the eQTLGen Consortium (https://www.eqtlgen.org) and was published in *Võsa*

*et al., 2021*. mQTL data originate from *Shin et al., 2014*, and are available at the Metabolomics GWAS Server (http://metabolomics.helmholtz-muenchen.de/gwas/). The HMDB was used to annotate metabolites and the v5.0 release from November 9, 2021, of the 'All proteins' file was downloaded to extract transcript-metabolite interactions (https://hmdb.ca/downloads). PubMed was used for the automated literature review (https://pubmed.ncbi.nlm.nih.gov). The UCSC Genome Browser (https://genome.ucsc.edu/) was used to visualize the *FADS* locus, while the GWAS Catalog was used to assess the number of reported GWAS signals in the region (https://www.ebi.ac.uk/gwas/). The STRING database was used for the enrichment analysis (https://string-db.org/). Produced data is available as *Supplementary file 1* and Source Data. Code used to perform analyses is freely available at https://github.com/eleporcu/Gene_Metab_Pheno; (*Porcu, 2022* copy archived at swh:1:rev:c6bff8d094e369ff0d399751fc85fcd5ea250134).

## Acknowledgements

Computations were carried out on the high-performance cluster of the Lausanne University Hospital (CHUV). This work was supported by funding from the Department of Computational Biology (ZK) and the Center for Integrative Genomics (AR) from the University of Lausanne, as well as funding from the Swiss National Science Foundation (310030-189147 to ZK and 31003A_182632 to AR).

## Additional information

### Funding

| Funder | Grant reference number | Author |
|---|---|---|
| Swiss National Science Foundation | 310030-189147 | Zoltán Kutalik |
| Swiss National Science Foundation | 31003A_182632 | Alexandre Reymond |

The funders had no role in study design, data collection and interpretation, or the decision to submit the work for publication.

### Author contributions

Chiara Auwerx, Writing – original draft, Investigation, Visualization; Marie C Sadler, Tristan Woh, Formal analysis, Writing – review and editing; Alexandre Reymond, Writing – review and editing; Zoltán Kutalik, Conceptualization, Writing – original draft; Eleonora Porcu, Conceptualization, Formal analysis, Writing – original draft

### Author ORCIDs

Chiara Auwerx ⓘ http://orcid.org/0000-0003-3613-8450
Marie C Sadler ⓘ http://orcid.org/0000-0002-2599-9207
Tristan Woh ⓘ http://orcid.org/0000-0001-6916-0174
Zoltán Kutalik ⓘ http://orcid.org/0000-0001-8285-7523
Eleonora Porcu ⓘ http://orcid.org/0000-0003-2878-7485

### Decision letter and Author response

Decision letter https://doi.org/10.7554/eLife.81097.sa1
Author response https://doi.org/10.7554/eLife.81097.sa2

## Additional files

### Supplementary files

• MDAR checklist

• Supplementary file 1. Supplementary tables. a. Annotation of metabolites measured in *Shin et al., 2014* with Human Metabolome Database (HMDB) identifiers. b. Significant transcript-to-metabolite causal effects (FDR 5%) identified through univariable Mendelian randomization. Both original

effects (ORIGINAL) and those after excluding outliers (N_outlier) are reported. The FDR column reports the adjusted p-value used to select significant associations (FDR ≤0.05). The HMDB and PubMed columns indicate the PMID of publications reporting a link between the tested transcript and metabolite, as identified per automated literature review, with '1*' indicating associations reported without referencing a specific publication. c. List of the 28 medically relevant phenotypes assessed in this study. d. Significant metabolite-to-phenotype causal effects (FDR 5%) identified through univariable metabolome-wide Mendelian randomization (MWMR). Both original effects (ORIGINAL) and those after excluding outliers (N_outlier) are reported. The FDR column reports the adjusted p-value used to select significant associations (FDR ≤0.05). e. Significant transcript-to-phenotype causal effects (FDR 5%) identified through univariable transcriptome-wide Mendelian randomization (TWMR). Both original effects (ORIGINAL) and those after excluding outliers (N_outlier) are reported. The FDR column reports the adjusted p-value used to select significant associations (FDR ≤0.05). f. Identified causal transcript-metabolite-phenotype triplets. Effect size and p-value for the transcript-to-metabolite, metabolite-to-phenotype, and transcript-to-phenotype relations among the 216 identified causal triplets, along with estimated direct and indirect effects. Rows colored in beige were identified by our automated literature review of transcript-to-metabolite pairs and were subjected to an automated literature review of the transcript-phenotype relation. The PubMed column reports the PMID of publications identified per automated literature review for the involved gene and phenotype (using the synonyms in PubMed_PHENO) after manual curation of abstracts to exclude findings in which search terms were used in an erroneous context. g. Metabolites integrating the effect of multiple transcripts. Twelve metabolites integrate the effect of multiple transcripts to in turn influence one or several phenotypes. Transcripts in bold in the same color are encoded by genes in close genomic proximity.

## Data availability

All data used in this study are publicly available. GWAS summary statistics for outcome traits measured in the UK Biobank originate from the Neale Lab (http://www.nealelab.is/uk-biobank/). eQTL data originated from the eQTLGen Consortium (https://www.eqtlgen.org) and was published in Vosa et al., 2021 [3]. mQTL data originate from Shin et al. 2014 [6], and are available at the Metabolomics GWAS Server (http://metabolomics.helmholtz-muenchen.de/gwas/). The Human Metabolome Database (HMDB) was used to annotate metabolites and the v5.0 release from 2021-11-09 of the "All proteins" file was downloaded to extract transcript-metabolite interactions (https://hmdb.ca/downloads). PubMed was used for the automated literature review (https://pubmed.ncbi.nlm.nih.gov). The UCSC Genome Browser (https://genome.ucsc.edu/) was used to visualize the *FADS* locus, while the GWAS Catalog was used to assess the number of reported GWAS signals in the region (https://www.ebi.ac.uk/gwas/). The STRING database was used for the enrichment analysis (https://string-db.org/). Produced data is available as Supplementary File 1 and Source Data. Code used to perform analyses is freely available at https://github.com/eleporcu/Gene_Metab_Pheno; (copy archived at swh:1:rev:c6bff8d094e369ff0d399751fc85fcd5ea250134).

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
