## [Editor Report]

The reviewers found that your article brings important new methods and insight for how to analyze large, complex, multi-omic datasets in order to highlight specific molecular hypotheses for follow-up validation. As realistic/affordable sample sizes for population studies with omics data have recently exploded in size, this has raised the clear need for new or adapted statistical methods for best exploiting the increased statistical power. This work is compelling and we believe it should be of general interest to biologists and biostatisticians, and of particular interest to those working on (or those who could work on) large cohorts.

---

## [Decision Letter]

**Decision letter after peer review:**

Thank you for submitting your article "Exploiting the mediating role of the metabolome to unravel transcript-to-phenotype associations" for consideration by *eLife*. Your article has been reviewed by 2 peer reviewers, including Evan Graehl Williams as Reviewing Editor and Reviewer #1, and the evaluation has been overseen by David James as the Senior Editor.

In brief, besides the reviewer-specific comments, there was a common thread regarding significances of the findings (both in the statistical sense and in the colloquial sense) and description of the method. I also apologize for the delay in the reviews; it was a huge challenge to get reviewers in July and August, and we potentially may consult a third reviewer for the revision.

*Reviewer #1 (Recommendations for the authors):*

Auwerx et al. have taken a new (or recent?) approach to mine large existing datasets of intermediary molecular data between GWAS and phenotype, with the aim of uncovering novel insight into the molecular mechanisms which lead a GWAS hit to have a phenotypic effect. The authors show that you can get additional insight by integrating multiple omics layers rather than analyzing only a single molecular type, including a handful of specific examples, e.g. that the effect of SNPs in ANKH on calcium are mediated by citrate. Such additional data is necessary because, as the authors' point out, while we have thousands of SNPs with significant impact on phenotypes of interest, we often don't know at all the mechanism, given that the majority of significant SNPs found through GWAS are in non-coding (and often intergenic) regions.

Scientific comments

(1) It's not quite clear to me whether the authors have *designed a new method* or have *applied a recently-developed method*. I follow that the general approach uses MR across multiple data types and then merges the results. I did look at the Github repo ("Gene_Metab_Pheno") but I'm not enough of a mathematical bioinformatician to really judge the novelty there, as to whether this is a new method or application of relatively recent other methods.

(2) The sample sizes are huge here since it is a meta-analysis, which means it's not quite clear to me how this method would scale to studies where there are not tens of thousands of individuals measured. It would be good to already have some general idea of what are the requirements in terms of power (sample size/target effect size / etc) for this approach – either added as a figure panel or two in this paper or just as text if the necessary parameters have already been explained elsewhere. I guess this comes back to the p >> n issue. I see for instance that the authors have 21k transcripts but they narrow it down to 7883 with {greater than or equal to} 3 IVs, used against their sample size of n = 7824 patients. Is this just convenient that n ~= p, or were the required parameters tweaked exactly so that you could reduce the number of transcripts down to more or less the sample size? You have a section on "Power Analysis" already which I appreciate, but reading through it again I can't tell what the minimum N would be. Or can I use an N = 500 I just have to use some other way to select only 500 (or whatever) transcripts? The power calculations go up to N = 90'000 which indeed is "state-of-the-art sample sizes" and would be unattainably large for all but the largest of international human cohorts, and perhaps a few large yeast or cell line studies.

(3) Related, I wasn't quite clear if the data from TwinsUK and KORA were analyzed separately, or if they were somehow merged.

(4) The authors mention "overall, 83% of the involved genes were causally influencing the level of a single metabolite, while TMEM258 and FADS2 affected 12 metabolites". Firstly I would mention that they affect 12 metabolites each just to be clear – although it's clearly annotated in Table S1. Second, and more relevantly, it would be nice to know in this table how many of these hits already are known in the literature. For instance I see the most significant hit is MAF1 with 5-oxoproline which doesn't immediately turn up anything (albeit I did not look very long). Hit #7, for SLC22A4 and isovalerylcarnitine does appear to be known in the literature ( http://www.metabolomix.com/rs272889-slc22a4-octn1-with-acylcarnitines/ ). Doing a decent literature analysis on all 257 of these hits in Table S1 might take a while (and/or require someone who knows how to do automated literature mining, which isn't trivial for this type of task especially given metabolite names like "X-12696") but even just some cherry-picking and/or (better) systematic analysis of the top 20-50 would be nice to see. It greatly improved my confidence in the findings when I just looked into two of your hits just now and found that one of them appears to be reasonably well known by literature.

(5) The authors mention that only 33% of the transcript-metabolite-phenotype triplets were observed by TWMR, but that 91% of those observed by all are in concordant directionality. I guess that means for those with concordant directionality, if no significant association was observed in one layer, there was just no cross-layer comparison? i.e. that the 91% "concordant directionality" are only checking among the 33% of hits by TWMR that were also observed both within and across layer?

(6) The authors mention no availability of trans-eQTL data; is that because the sample size is not sufficient to do a proper trans-eQTL analysis, or is it because of data privacy and an inability to get the precomputed trans-eQTLs from the eQTLGen Consortium due to some potential conflict (GDPR?).

(7) Back to Table S1: There are 257 hits here, with 191 genes (out of 7883) and 154 metabolites (out of 453), meaning that 299 metabolites didn't even have one significant association?

*Reviewer #2 (Recommendations for the authors):*

1. In general, the methodological approach should be explained in more detail. The study presents a novel MR-based approach and thus, the discussion of the method basics should be elaborated. This pertains to the following points:

– In the definition of causal effects from molecular traits on the outcome (eq. (1)), the symbol β' is not explained. The method used for standardization of effect sizes is not elaborated.

– For the multivariable MR analysis (equation not numbered), no causal effect statistic is specified. As discussed in the public review, no significance testing has been performed, but we strongly urge the authors to do so or give reasons for this lack of testing in the case of causal triplets.

– In the description of the simulation analyses, it should be made more clear how the definition of βGWAS as a function of βmQTL is reached.

– The authors should include a thorough discussion and a description of the method employed to evaluate the comparison between total and direct transcript-phenotype effects in Figure 2B.

2. There is some apparent inconsistency in the interpretation of results from the empirical study: The abstract states that 67 transcript-phenotype effects present within the 206 causal triplets were missed in a TWMR-only analyses, while the paragraph "Power Analysis" states that only 67 triplets showed a significant effect estimated by TWMR. The text relating to Figure 2B states that 79 % (~163) of the triplets were estimated to be due to direct effects by the transcript only. This should be presented in a more consistent and clear manner.

3. Plots for the distribution of the number of IVs per transcript/metabolite should be included (as supplemental figures).

4. It would be interesting to discuss potential modes of mediation between transcripts, metabolites and phenotypes. For example, individual metabolites might integrate the effects of several transcripts. This would not result in triplets, but ultimately, networks of mediation as shown for the TMEM258, FADS1, FADS2 case. How would such cases be treated in a statistically sound manner?

---

## [Author Response]

Reviewer #1 (Recommendations for the authors):Auwerx et al. have taken a new (or recent?) approach to mine large existing datasets of intermediary molecular data between GWAS and phenotype, with the aim of uncovering novel insight into the molecular mechanisms which lead a GWAS hit to have a phenotypic effect. The authors show that you can get additional insight by integrating multiple omics layers rather than analyzing only a single molecular type, including a handful of specific examples, e.g. that the effect of SNPs in ANKH on calcium are mediated by citrate. Such additional data is necessary because, as the authors' point out, while we have thousands of SNPs with significant impact on phenotypes of interest, we often don't know at all the mechanism, given that the majority of significant SNPs found through GWAS are in non-coding (and often intergenic) regions.Scientific comments(1) It's not quite clear to me whether the authors have designed a new method or have applied a recently-developed method. I follow that the general approach uses MR across multiple data types and then merges the results. I did look at the Github repo ("Gene_Metab_Pheno") but I'm not enough of a mathematical bioinformatician to really judge the novelty there, as to whether this is a new method or application of relatively recent other methods.

While the core Mendelian randomization (MR) and mediation analysis methodology used in this study is well-established, the novelty of the manuscript lies in (i) the strategy that sequential application of univariable and multivariable MR analyses across the different omics layers (i.e., genome, transcriptome, metabolome, phenome) can improve discovery power and (ii) the resulting key message that by incorporating metabolomic data to genetic studies, it becomes possible to gain functional and targetable insights into the pathways linking genetic variation to phenotypic diversity. To clarify this point, we have modified the Introduction of our revised manuscript:

“Here, we develop a framework based on established MR methodology that hypothesizes a mediating role of the metabolome in the transcript-to-phenotype axis. Specifically, our integrative MR analysis combines summary-level multi-omics (i.e., GWAS, eQTL, and mQTL) data to compute the indirect effect of gene expression on complex traits mediated by metabolites in three steps (Figure 1).”

(2) The sample sizes are huge here since it is a meta-analysis, which means it's not quite clear to me how this method would scale to studies where there are not tens of thousands of individuals measured. It would be good to already have some general idea of what are the requirements in terms of power (sample size/target effect size / etc) for this approach – either added as a figure panel or two in this paper or just as text if the necessary parameters have already been explained elsewhere. I guess this comes back to the p >> n issue. I see for instance that the authors have 21k transcripts but they narrow it down to 7883 with {greater than or equal to} 3 IVs, used against their sample size of n = 7824 patients. Is this just convenient that n ~= p, or were the required parameters tweaked exactly so that you could reduce the number of transcripts down to more or less the sample size?

Requiring at least 3 IVs is common practice in the field of MR as it allows to estimate the causal estimate more robustly, as previously described (PMID: 31341166). In general, the more instruments, the less outliers can bias results. If the InSIDE assumption holds the MR estimates are asymptotically unbiased, hence more instruments reduce bias. Finally, more instruments also increase discovery power. Hence, the fact that the number of transcripts is reduced to the sample size of the study is pure coincidence. Of note, while this reduces the multiple testing burden, it also means that we potentially miss out on true causal effects of transcripts that cannot be instrumented properly.

You have a section on "Power Analysis" already which I appreciate, but reading through it again I can't tell what the minimum N would be. Or can I use an N = 500 I just have to use some other way to select only 500 (or whatever) transcripts? The power calculations go up to N = 90'000 which indeed is "state-of-the-art sample sizes" and would be unattainably large for all but the largest of international human cohorts, and perhaps a few large yeast or cell line studies.

While we are hopeful that in the future, ever larger multi-omics datasets will become available to the research community (e.g., initiatives similar to the NMR metabolite measurement in >120,000 UK Biobank participants; https://biobank.ctsu.ox.ac.uk/crystal/label.cgi?id=220), we agree that the majority of currently available molecular datasets have smaller sample sizes than the one used in this study. To address this comment, we ran power analyses simulating mQTLs studies performed on 1000, 2000, and 4000 individuals. Following the previously observed trend, the gain in power of mediation analysis over TWMR decreases with decreasing N and is quasi inexistant with sample sizes smaller than 2,000. Hence, the reviewer’s point is well taken, the mediator sample size is crucial for our main conclusions. These additional results are detailed in the revised manuscript, along with Figure 4 —figure supplement 2:

“Repeating the simulations with a mQTL sample size of 90,000, nearing state-of-the-art sample sizes [7], we observe a strong shift in the above-described trends (Figure 4B). Specifically, when the effect of the transcript on the phenotype is dominated by the effect mediated by the metabolite (ρ<0.3 and ρ>1.7), mediation analysis has more power than TWMR when σ>0.2. For larger proportions of direct effect, TWMR has increased power the more σ differs from 1. In line with the increased power of mediation analysis with larger mQTL datasets, the gain in power of mediation analysis over TWMR decreases with decreasing mQTL dataset sample sizes (ranging between N=1,000 and N=4,000; Figure 4 —figure supplement 2), indicating that our approach is dependent on large sample sizes to reach its full potential.”

(3) Related, I wasn't quite clear if the data from TwinsUK and KORA were analyzed separately, or if they were somehow merged.

Data from KORA and TwinsUK have previously been analyzed, as described in Shin *et al.*, 2014 (PMID: 24816252). Specifically, genome-wide association analyses were performed separately in KORA and TwinsUK for all 486 metabolite concentrations present in both studies after quality control steps. Association results were then combined using inverse variance meta-analysis based on effect size estimates and standard errors, adjusting for genomic control. Our study solely uses the publicly available meta-analysis summary statistics provided by Shin *et al.*, 2014 for 453 of these metabolites.

We would also like to highlight that based on comments by Reviewer #1 and #2, the revised manuscript now focuses on a restricted set of 242 metabolites for which we were able to identify a Human Metabolome Database (HMDB) identifier through manual annotation. This approach is motivated by only analyzing metabolites whose potential associations can be followed up.

The current version of the manuscript reads as follows:

Results: “Summary statistics for cis-eQTLs stem from the eQTLGen Consortium meta-analysis of 19,942 transcripts in 31,684 individuals [3], while summary statistics for mQTLs originate from a meta-analysis of 453 metabolites in 7,824 individuals from two independent European cohorts: TwinsUK (N = 6,056) and KORA (N = 1,768). After selecting SNPs included in both the eQTL and mQTL studies, our analysis was restricted to 7,884 transcripts with ≥ 3 instrumental variables (IVs) (see Methods, Figure 1 —figure supplement 1A) and 242 metabolites with an identifier in The Human Metabolome Database (HMDB) (see Methods, Supplementary File 1a).”

Methods: “mQTL data originate from Shin et al., 2014 which used ultra-high performance liquid chromatography-tandem mass spectrometry (UPLC-MS/MS) to measure 486 whole blood metabolites in 7,824 European individuals. Association analyses were carried out on ~2.1 million SNPs and are available for 453 metabolites at the Metabolomics GWAS Server (http://metabolomics.helmholtz-muenchen.de/gwas/). Among these metabolites, 242 were manually annotated with Human Metabolome Database (HMDB) identifiers (Supplementary File 1a) and used in this study.”

(4) The authors mention "overall, 83% of the involved genes were causally influencing the level of a single metabolite, while TMEM258 and FADS2 affected 12 metabolites". Firstly I would mention that they affect 12 metabolites each just to be clear – although it's clearly annotated in Table S1.

This has been clarified in the revised manuscript:

“Most involved genes (86%; 83/96) were causally influencing the level of a single metabolite, with some notable exceptions acting as mQTL hubs, such as TMEM258 and FADS2 both affecting the same 11 metabolites, followed by FADS1 affecting a subset of 6 metabolites.”

Second, and more relevantly, it would be nice to know in this table how many of these hits already are known in the literature. For instance I see the most significant hit is MAF1 with 5-oxoproline which doesn't immediately turn up anything (albeit I did not look very long). Hit #7, for SLC22A4 and isovalerylcarnitine does appear to be known in the literature (http://www.metabolomix.com/rs272889-slc22a4-octn1-with-acylcarnitines/ ). Doing a decent literature analysis on all 257 of these hits in Table S1 might take a while (and/or require someone who knows how to do automated literature mining, which isn't trivial for this type of task especially given metabolite names like "X-12696") but even just some cherry-picking and/or (better) systematic analysis of the top 20-50 would be nice to see. It greatly improved my confidence in the findings when I just looked into two of your hits just now and found that one of them appears to be reasonably well known by literature.

This is a great suggestion which led us to conduct a more detailed follow-up investigation. Having a rough estimation of the number of associations that have previously been described in the literature would indeed be of great interest. Due to the large number of identified associations, performing a thorough, manual investigation of each of the identified associations is hardly manageable. As an alternative, we used an automated literature review strategy that we briefly describe below.

First, we reduced the number of investigated metabolites to 242 interpretable compounds for which we could identify an HMDB identifier through manual annotation (see reply to scientific comment #3). Second, using this annotation, we identified transcript-to-metabolite effects for which an interaction between the encoded protein and metabolite was reported in the HMDB. In parallel, we performed an automated literature review in PubMed for all identified transcript-to-metabolite effects. Next, for the 37 causal transcript-metabolite-phenotype triplets for which either the HMDB or the PubMed search for the transcript-to-metabolite yielded at least one results, we (i) added a third term to the PubMed search relating to the associated phenotype and (ii) replaced the metabolite term by the phenotype term to determine whether the transcript-to-phenotype analysis had previously been reported. Owing to the low specificity of the phenotype search term (e.g., “creatinine” appearing in “X mmol/mol creatinine”), we manually reviewed the abstracts of retained publications, excluding findings in which the search terms were used in an unintended way. We describe the methods and results of these analyses in detail in the revised manuscript – along with a discussion of the limitations of our approach – and provide the PMIDs of our automated search in the Supplementary File 1b:

Methods:

“Omics and traits summary statistics

[…] mQTL data originate from Shin et al. 2014, which used ultra-high performance liquid chromatography-tandem mass spectrometry (UPLC-MS/MS) to measure 486 whole blood metabolites in 7,824 European individuals. Association analyses were carried out on ~2.1 million SNPs and are available for 453 metabolites at the Metabolomics GWAS Server (http://metabolomics.helmholtz-muenchen.de/gwas/). Among these metabolites, 242 were manually annotated with Human Metabolome Database (HMDB) identifiers (Supplementary File 1a) and used in this study. GWAS summary statistics for 28 outcome traits measured in the UK Biobank (UKB) originate from the Neale Lab (http://www.nealelab.is/uk-biobank/). Protein interactions with metabolites were downloaded from HMDB v5.0 (https://hmdb.ca/downloads/hmdb_proteins.xml) and were used to annotate transcript-metabolites associations.

Automated literature review

An automated literature review of all transcript-metabolites associations (Supplementary File 1b) was conducted in PubMed (20/09/2022) following the scheme:(Gene[Title/Abstract]) AND ((MetHMDB[Title/Abstract]) OR (MetShin[Title/Abstract])) With Gene being the name of the gene whose transcript is involved in the association, MetHMDB the involved metabolite’s common name on HMDB, and MetShin the involved metabolite’s name as reported in Shin et al., [6]. Returned PubMed identifiers were retrieved (Supplementary File 1b).

For transcript-metabolite associations involved in a causal triplet and for which the transcript-metabolites returned at least one publication (Supplementary File 1f), the search was extend by (i) adding a further search term for the trait (i.e., AND (trait[Title/Abstract])) and (ii) substituting the metabolite term for the trait term. Returned PubMed identifiers were retrieved and corresponding abstracts were manually curated to exclude abstracts in which the search terms were used in a meaning other than the intended one (Supplementary File 1f).”

Results:

“Mapping the transcriptome onto the metabolome

[…] While only 5 (3.8%) of the 133 associations were reported in HMDB, an automated literature review (see Methods) identified a match for 22 (16.5%) of the identified transcript-metabolite pairs (Supplementary File 1b).

Mapping metabolome-mediated effects of the transcriptome onto complex phenotypes

[…] Among the 37 triplets for which the transcript and metabolite had previously been linked through automated literature review, none remained after incorporating a third term for the phenotype in the search and manually removing abstracts for which the search terms were used in an erroneous context. Relaxing the search criteria by omitting the metabolite term, 13/37 (35%) triplets returned at least one match for the gene-trait association.”

Discussion:

“Our automated literature review indicates that while some detected associations were previously reported, a large fraction, especially among the triplets, appears to be novel. It should be noted that the number of previously reported associations is likely underestimated as our approach does not account for all synonyms of a given feature and requires the terms to appear in the title or abstract of the publication. This makes it more likely for hypothesis-driven studies, inherently biased towards well-studied genes and metabolites, to be identified. Conversely, high-throughput, hypothesis-free studies that report the given association in a supplemental table are likely to be missed. Furthermore, due to its automated nature, our search is context-blind, so that some of the identified studies might report negative results, associations only under specific conditions (e.g., different organisms, experimental settings), or usage of the search term with a different meaning. To attenuate the latter, we also performed manual review of the retained abstracts for transcript-to-phenotype searches. While flawed, this rough estimate of the amount of existing evidence supporting our findings can be interpreted in combination with other lines of evidence. For instance, […].”

(5) The authors mention that only 33% of the transcript-metabolite-phenotype triplets were observed by TWMR, but that 91% of those observed by all are in concordant directionality. I guess that means for those with concordant directionality, if no significant association was observed in one layer, there was just no cross-layer comparison? i.e. that the 91% "concordant directionality" are only checking among the 33% of hits by TWMR that were also observed both within and across layer?

In response to Reviewer #2’s comments, the current version of the manuscript uses more stringent significant thresholds. At FDR 5%, we now observe 216 transcript-metabolite-phenotype triplets. Among these 216 triplets, 90 (42%) had a significant transcript-to-phenotype TWMR effect. Among these 90 triplets, 84 (93%) showed a directionally concordant effect between the transcript-to-phenotype, transcript-to-metabolite, and metabolite-to-phenotype estimates, meaning that the sign of the product of the transcript-to-metabolite and metabolite-to-phenotype effects equals the sign of the transcript-to-phenotype effect. This is now clarified in the Discussion:

“[…] among the 90 signals that were also identified through TWMR, 93% showed a directionally concordant effect between the transcript-to-phenotype, transcript-to-metabolite, and metabolite-to-phenotype estimates (i.e., sign of the product of the transcript-to-metabolite and metabolite-to-phenotype effects agrees with the sign of the transcript-to-phenotype effect).”

(6) The authors mention no availability of trans-eQTL data; is that because the sample size is not sufficient to do a proper trans-eQTL analysis, or is it because of data privacy and an inability to get the precomputed trans-eQTLs from the eQTLGen Consortium due to some potential conflict (GDPR?).

The trans-eQTLs data from eQTLGen Consortium are restricted to ~10,000 SNPs reported to be associated with complex traits in GWAS catalog due to the massive size of this type of data. There are current efforts within the eQTLGen Consortium to meta-analyse trans-eQTL summary statistics genome-wide, but these data will not be available before Q3 of 2023. Currently available data is unfortunately not sufficient to investigate reverse causality on metabolites and gene expression, as specified in the manuscript’s Discussion:

“Due to the lack of genome-wide trans-eQTL association summary statistics, our method does not investigate reverse causality on metabolites and gene expression […].”

(7) Back to Table S1: There are 257 hits here, with 191 genes (out of 7883) and 154 metabolites (out of 453), meaning that 299 metabolites didn't even have one significant association?

This is correct. In the current version of the manuscript and as explained in the reply to previous points, we restricted our analysis to 242 metabolites manually annotated with HMDB identifiers. In Supplementary File 1b, we report 133 significant (FDR < 5%) transcript-to-metabolite effects involving 75 unique metabolites, so that in case of 167 metabolites we do not have evidence for them being impacted by any of the 7,824 tested genes. It is interesting to notice that despite changes in our analysis (i.e., restrict to interpretable metabolites, more stringent criterion for significance), the proportion of metabolites that are not affected by any transcripts remains similar (previously: 299/453 = 66%; currently: 167/242 = 69%). One potential explanation is lack of power, given the relatively small sample size of the mQTL dataset (< 8,000), as compared to GWASs for other complex traits (> hundreds of thousands), that prevents detection of associations for a large fraction of metabolites. Alternatively, these metabolites might be influenced by transcripts that cannot be instrumented, making it impossible to detect association.

Reviewer #2 (Recommendations for the authors):1. In general, the methodological approach should be explained in more detail. The study presents a novel MR-based approach and thus, the discussion of the method basics should be elaborated. This pertains to the following points:– In the definition of causal effects from molecular traits on the outcome (equation (1)), the symbol β' is not explained. The method used for standardization of effect sizes is not elaborated.

We clarified this in the current version of the manuscript, which now reads as follows:

“Here, β is a vector of length n containing the standardized effect size of n independent SNPs on the gene/metabolite, derived from eQTL/mQTL studies, with β′ being the transpose of β. γ is a vector of length n containing the standardized effect size of each SNP on the outcome. C is the pairwise LD matrix between the n SNPs. The standardized effect sizes for molecular and outcome GWASs were obtained from Z-score of summary statistics standardized by the square root of the sample size to be on the same standard deviation scale.”

– For the multivariable MR analysis (equation not numbered), no causal effect statistic is specified. As discussed in the public review, no significance testing has been performed, but we strongly urge the authors to do so or give reasons for this lack of testing in the case of causal triplets.

The equation has now been labeled (6). As specified in the Methods, we used multivariable MR (MVMR) only to estimate the direct effect of gene expression on the phenotype. We do not test the significance of this estimate as we only compare it with the total effect estimated by TWMR regardless its significance. Regarding significance testing for other sections of the manuscript, please see extensive reply to major concern #1 and #2.

– In the description of the simulation analyses, it should be made more clear how the definition of βGWAS as a function of βmQTL is reached.

As described in the Methods, to estimate the effect of the metabolite on the phenotype, we considered a metabolite with heritability hM2=0.04 (corresponding to the median of h2 in the KORA+TwinsUK mQTL data) and NIVs=5 (corresponding to the median number of IVs used in MWMR analyses). Effect size βmQTL are from a normal distribution βmQTL∼N(0,hM2NIVs). These quantities allowed to define βGWAS as βGWAS= αTP∗(1−ρ)/σ∗ βmQTL+ εP, where εP∼N(0,1NGWAS).

– The authors should include a thorough discussion and a description of the method employed to evaluate the comparison between total and direct transcript-phenotype effects in Figure 2B.

We have now added a description of the comparison of the direct and total effects in the Methods:

“The proportion of direct effect (ρ) is calculated by regressing direct effects (αd) on total effects (αTP) and then correcting for regression dilution bias:

ρcorrected= ρ1−∑(SE(αTP))2∑αTP2 ”

It is crucial that due to lack of power, we do not compare individual direct-vs-total effects but only assess overall direct-vs-total effects across all transcript-phenotype pairs. The central message of the paper is not the difference between the two, but the difference in detection power between the two strategies, as we do illustrate in Figure 4.

2. There is some apparent inconsistency in the interpretation of results from the empirical study: The abstract states that 67 transcript-phenotype effects present within the 206 causal triplets were missed in a TWMR-only analyses, while the paragraph "Power Analysis" states that only 67 triplets showed a significant effect estimated by TWMR.

We apologize for this inconsistency, which has been clarified. The revised sections now read:

Abstract: “We identified 216 transcript-metabolite-trait causal triplets involving 26 medically relevant phenotypes. Among these associations, 58% were missed by classical transcriptome-wide MR […].”

Results: “Importantly, only 42% (90/216) of the causal triplets showed a significant total transcript-to-phenotype effect (i.e., estimated by TWMR), suggesting that the method lacks power under current settings.”

The text relating to Figure 2B states that 79 % (~163) of the triplets were estimated to be due to direct effects by the transcript only. This should be presented in a more consistent and clear manner.

In Figure 2B, we plot the direct (αd; y-axis) and total (αTP; x-axis) effects for the 216 transcript-metabolite-trait causal triplets. Regressing direct effects on total effects, we observed that 77% of the transcript effect on the phenotype is direct and thus not mediated by the metabolites. This is explained in the Results and Methods of the manuscript:

Results:

“Regressing direct effects (αd) on total effects (αTP) and accounting for regression dilution bias (see Methods; Figure 2A), it was estimated that 77% [95% CI: 70%-85%] of the transcript effect on the phenotype was direct and thus not mediated by the metabolites (Figure 2B).”

Methods:

“The proportion of direct effect (ρ) is calculated by regressing direct effects (αd) on total effects (αTP) and then correcting for regression dilution bias:

ρcorrected= ρ1−∑(SE(αTP))2∑αTP2 ”

3. Plots for the distribution of the number of IVs per transcript/metabolite should be included (as supplemental figures).

We now include Figure 1 —figure supplement 1 to illustrate this.

We furthermore refer to these plots in the Results section:

“Mapping the transcriptome onto the metabolome

[…] After selecting SNPs included in both the eQTL and mQTL studies, our analysis was restricted to 7,884 transcripts with ≥ 3 instrumental variables (IVs) (see Methods, Figure 1 —figure supplement 1A) […].

Mapping the metabolome onto complex phenotypes

Univariable metabolome-wide MR (MWMR; Figure 1A) was used to identify causal relationships between 48 metabolites with ≥ 3 IVs and 28 complex phenotypes (Figure 1 —figure supplement 1B). […]

Mapping the transcriptome onto complex phenotypes

We applied univariable transcriptome-wide MR (TWMR [12]; Figure 1B) to identify associations between expression levels of 10,435 transcripts from the eQTLGen Consortium with ≥ 3 IVs (Figure 1 —figure supplement 1C) measured in both exposure and outcome datasets and the same 28 UKB phenotypes described in the previous section (Supplementary File 1c)[…].”

4. It would be interesting to discuss potential modes of mediation between transcripts, metabolites and phenotypes. For example, individual metabolites might integrate the effects of several transcripts. This would not result in triplets, but ultimately, networks of mediation as shown for the TMEM258, FADS1, FADS2 case. How would such cases be treated in a statistically sound manner?

This is a very interesting idea and we believe that the field is increasingly interested in modelling such complex biological network. The situation where a single metabolite mediates the effect of multiple transcripts only needs a systematic search for metabolites repeatedly occurring in triplets. We identified 12 such metabolites (see Table integrated in revised manuscript as Supplementary File 1g). Nine out of 12 metabolites integrate effect of multiple transcripts to in turn affects multiple phenotypes, as illustrated in the last column of the Table. Of note 8/12 cases, the metabolites integrate the effect of transcript whose genes are in close genomic proximity (shown as bold transcripts in the same color). We already made this observation in the original manuscript and discussed this limitation:

“Owing to linkage disequilibrium and regulatory variants affecting multiple genes, transcripts from adjacent genes might appear to be involved in the same signals, as exemplified with the TMEM258/FADS1/FADS2 locus (Figure 3). While literature supports the role of the FADS genes, one cannot exclude a role for TMEM258, nor disentangle the specific function of FADS1 and FADS2.”

Investigating more complex scenarios where multiple correlated metabolites mediate the effect of a single transcript would require more advanced modelling techniques that go beyond the scope of this study. We strengthened the Discussion on the topic in the revised manuscript which now reads as follows:

“Another consideration is that complex phenotypes can have a stronger impact on gene expression than the opposite [15]. Due to the lack of genome-wide trans-eQTL association summary statistics, our method does not investigate reverse causality on metabolites and gene expression, nor the role of metabolites as regulators of gene expression. Metabolites might also integrate the effect of several transcripts (i.e., multiple transcripts causally impact the levels of the same metabolite) before affecting complex phenotypes (Supplementary File 1g) or multiple metabolites may jointly mediate the impact of a single transcript. Modelling the latter phenomenon, which is beyond the scope of our current work, requires the development of Structural Equation Models accounting for such effects and will eventually lead to a more comprehensive modelling of causal relations in complex biological networks, nuancing the interpretation of the molecular mechanisms shaping complex traits.”